# Design and Synthesis of a Cyclic Double-Grafted Polymer Using Active Ester Chemistry and Click Chemistry via A “Grafting onto” Method

**DOI:** 10.3390/polym11020240

**Published:** 2019-02-01

**Authors:** Meng Liu, Lu Yin, Shuangshuang Zhang, Zhengbiao Zhang, Wei Zhang, Xiulin Zhu

**Affiliations:** 1Suzhou Key Laboratory of Macromolecular Design and Precision Synthesis, Jiangsu Key Laboratory of Advanced Functional Polymer Design and Application, State and Local Joint Engineering Laboratory for Novel Functional Polymeric Materials, College of Chemistry, Chemical Engineering and Materials Science, Soochow University, Suzhou 215123, China; 20164209036@stu.suda.edu.cn (M.L.); yinlu0701@163.com (L.Y.); zhengbiaozhang@suda.edu.cn (Z.Z.); xlzhu@suda.edu.cn (X.Z.); 2Global Institute of Software Technology, No 5. Qingshan Road, Suzhou National Hi-Tech District, Suzhou 215163, China

**Keywords:** cyclic double-grafted polymer, topological architecture, active ester chemistry, “grafting onto” approach

## Abstract

Combing active ester chemistry and click chemistry, a cyclic double-grafted polymer was successfully demonstrated via a “grafting onto” method. Using active ester chemistry as post-functionalized modification approach, cyclic backbone (*c*-P2) was synthesized by reacting propargyl amine with cyclic precursor (poly(pentafluorophenyl 4-vinylbenzoate), *c*-PPF4VB_6.5k_). Hydroxyl-containing polymer double-chain (*l*-PS-PhOH) was prepared by reacting azide-functionalized polystyrene (*l*-PSN_3_) with 3,5-bis(propynyloxy)phenyl methanol, and further modified by azide group to generate azide-containing polymer double-chain (*l*-PS-PhN_3_). The cyclic backbone (*c*-P2) was then coupled with azide-containing polymer double-chain (*l*-PS-PhN_3_) via CuAAC reaction to construct a novel cyclic double-grafted polymer (*c*-P2-*g*-Ph-PS). This research realized diversity and complexity of side chains on cyclic-grafted polymers, and this cyclic double-grafted polymer (*c*-P2-*g*-Ph-PS) still exhibited narrow molecular weight distribution (*M*_w_/*M*_n_ < 1.10).

## 1. Introduction

Grafted polymers, in conjunction with conventional linear polymers, have fascinating topological macromolecular structure with various side chains along their main backbones. They exhibit unique and remarkable properties by controlling chemical components of main backbones or side chains, the length of side chains, and grafting density [1,2]. Grafted polymers have been thoroughly probed as the precursors of drug-delivery materials [3], biosensors [4], nanowires [5] and nanotubes [6]. According to the structure of polymeric main backbones, grafted polymers can be roughly divided into linear-, star-, cyclic-, dendritic-, hyperbranched-grafted polymers and so on. With continuous development of new synthesis technologies, novel grafted polymers have exhibited more and more different properties and exploration for new strategies to construct novel grafting polymers have never stopped [7,8,9,10,11,12,13,14,15].

Comparing linear counterparts, cyclic polymers without any terminals, as another kind of topological architecture, demonstrate remarkable and unique properties, such as a smaller hydrodynamic volume, lower intrinsic viscosity, a higher glass transition temperature, higher rate of crystallization and a higher density [16,17,18,19,20,21,22]. In general, the research of cyclic polymers can be mainly divided into three parts: (1) constructing complex cyclic topological structure; (2) exploring new synthetic methods and strategies of cyclic polymers; (3) comparing the performance difference between cyclic polymers and linear counterparts [23,24,25,26,27]. Recent studies related to cyclic polymers have displayed that partial bio-materials containing cyclic structures could reduce cytotoxicity [28], increase transfection efficiency [29] or improve drug loading and releasing capacity [30,31].

Cyclic-grafted polymers consisting of one cyclic backbone and various side chains, as one kind of grafted polymers, have potential applications in biomaterials [32,33,34]. There are three main categories for producing cyclic-grafted polymers: (1) “grafting through” approach, the polymerization of cyclic-macromonomers [35]; (2) “grafting from” approach, the growth of side chains from a cyclic-macroinitiator backbone [36,37,38,39,40,41,42]; (3) “grafting onto” approach, the addition of ready-made polymeric chains to a cyclic backbone by high-effective chemical reactions or supramolecular assembly, such as esterification reaction [43], click chemistry [44,45,46,47,48], active ester chemistry [49,50], Suzuki coupling reaction [51] and metallo-supramolecular interactions [52]. In the “grafting onto” approach, cyclic backbone and grafting chains can be independently synthesized and characterized. Diversity and complexity of side chains probably contribute to constructing functionalized cyclic topologies and exploring their potential applications.

In this work, combing active ester chemistry and click chemistry, we constructed a cyclic double-grafted polymer successfully via the “grafting onto” approach as shown in Scheme 1. Using active ester chemistry, a cyclic backbone (*c*-P2) was synthesized by post-functionalized modification by reacting propargyl amine with cyclic precursor (poly(pentafluorophenyl 4-vinylbenzoate), *c*-PPF4VB_6.5k_) (Appendix A). Additionally, hydroxyl-containing polymer double-chain (*l*-PS-PhOH) was prepared by reacting azide-functionalized polystyrene (*l*-PSN_3_) with 3,5-bis(propynyloxy)phenyl methanol, and further modified by azide group to generate azide-containing polymer double-chain (*l*-PS-PhN_3_). The cyclic backbone (*c*-P2) was then coupled with prepared polymer double-chain (*l*-PS-PhN_3_) using CuAAC reaction to successfully construct a novel cyclic double-grafted polymer (*c*-P2-*g*-Ph-PS). This research realized diversity and complexity of side chains on cyclic-grafted polymers, and this cyclic double-grafted polymer (*c*-P2-*g*-Ph-PS) still exhibited narrow molecular weight distribution.

## 2. Materials and Methods 

### 2.1. Materials

3,5-Dihydroxybenzyl alcohol (HWRK CHEM, Beijing, China, 98%), 3-(trimethylsilyl)propargyl bromide (Energy Chemical, Shanghai, China, 97%), 18-crown-6 (Sinopharm Chemical Reagent, Suzhou, China, CP), diphenyl azidophosphate (DPPA) (Alfa Aesar, Shanghai, China, 97%), 1,8-diazabicylco[5.4.0]undec-7-ene (DBU) (Tokyo Chemical Industry Co., Ltd., 98%), pentamethyldiethylenetriamine (PMDETA) (Energy Chemical, Shanghai, China, 99%) and copper(0) powder (Sinopharm Chemical Reagent, Suzhou, China, 99.9%) were purchased and used as received. Copper(I) bromide (CuBr) (Sinopharm Chemical Reagent, Suzhou, China, 99%) was washed by the mixture solution (acetic acid/deionized water, *v*/*v* = 5/95) and anhydrous ethanol many times, and then dried in a vacuum. The solvents including acetone, dichloromethane (CH_2_Cl_2_), hydrochloric acid (HCl), toluene, ethyl acetate (EA), petroleum ether (PE), tetrahydrofuran (THF), dimethylformamide (DMF) and methanol (MeOH) were used as received without any purification process.

### 2.2. Characterizations

All the ^1^H NMR and ^13^C NMR spectra were measured on a Bruker (300 MHz) Nuclear Magnetic Resonance spectrometer (Bruker, USA). All the average molecular weights (*M*_n_) and molecular weight distributions (*M*_w_/*M*_n_) were measured by TOSOH HLC-8320 size exclusion chromatography (SEC, Tosoh Corporation, Japan). The recycling preparative HPLC Mode LC-9260NEXT (often called as Prep-SEC, Tosoh Corporation, Japan) was utilized to purify crude polymers. A Bruker TENSOR-27 FT-IR spectrometer was utilized to measure FT-IR spectra (Bruker, USA). Matrix assisted laser desorption ionization/time of flight mass spectra (MALDI TOF MS) (Bruker, USA) were gained by using an UltrafleXtreme MALDI TOF mass spectrometer. The UV-light resource was considered using one low-pressure lamp purchased from Beijing China Education Au-light Co. Ltd (CEL-LPH120-254, 120 W, Beijing, China). All the parameters and measure conditions of these spectrometers are shown in detail in the Appendix A.

### 2.3. Synthesis of 3,5-bis(propargyloxy)benzyl Alcohol

3,5-Dihydroxybenzyl alcohol (1.4 g, 10 mmol), 3-(trimethylsilyl)propargyl bromide (4.2 g, 22 mmol), 18-crown-6 (264 mg, 1 mmol), K_2_CO_3_ (3.45 g, 25 mmol) and acetone (80 mL) were added into a round-bottom flask in nitrogen atmosphere. This mixture solution was placed in oil-bath at 80 °C for 48 h. The mixed solution was filtered to remove indissoluble solid, and the filtrate was then concentrated. The solution was extracted with CH_2_Cl_2_ for three times and washed by 1 mol/L HCl and brine. The collected organic phase was dried using MgSO_4_. After evaporating the solution, the crude product was purified by silica gel chromatography (eluent: petroleum ether/ethyl acetate = 4/1) to get a white solid (1.2 g, yield: 57.3%). ^1^H NMR (CDCl_3_, 300 MHz, ppm, Appendix A): 6.63 (a, 2H), 6.54 (b, 1H), 4.67 (c, 6H), 2.53 (d, 2H), 1.70 (e, 1H). ^13^C NMR: (CDCl_3_, 75 MHz, ppm): 158.92, 143.70, 106.31, 101.58, 78.49, 75.81, 65.14, 56.01.

### 2.4. Synthesis of Hydroxyl-Containing Polymer Double-Chain (l-PS-PhOH)

3,5-Bis(propargyloxy)benzyl alcohol (10.81 g, 0.05 mmol), azide-functional polystyrene (*l*-PS-N_3_, 257.5 mg, 0.103 mmol), PMDETA (17.84 mg, 0.103 mmol), toluene (5 mL) and a magnetic stirrer were added into a 10 mL ampoule in nitrogen atmosphere. CuBr (14.8 mg, 0.103 mmol) and Cu (3.18 mg, 0.05 mmol) were added into above ampoule. The mixture solution was stirred at ambient temperature for 1.5 h in nitrogen atmosphere. After that, the polymer was precipitated in anhydrous methanol and dried in a vacuum (249.4 mg, yield: 92.9%). The crude polymer double-chain was purified by Prep-SEC to get hydroxyl-containing polymer double-chain (*l*-PS-PhOH, *M*_n,SEC_ = 5000 g/mol, *M*_w_/*M*_n_ = 1.04).

### 2.5. Synthesis of Azide-Containing Polymer Double-Chain (l-PS-PhN_3_)

Hydroxyl-containing polymer double-chain (*l*-PS-PhOH, 100 mg, 0.02 mmol) was dissolved in DMF (1 mL) and put in an ampoule (5 mL) containing a magnetic stirrer in nitrogen atmosphere. The ampoule was wrapped in aluminum foil to avoid light. DPPA (110.08 mg, 0.4 mmol) and DBU (60.90 mg, 0.4 mmol) were added into the ampoule under nitrogen atmosphere. The ampoule was placed in an oil-bath at 80 °C for 24 h. The mixed solution was purified by passing through a short Al_2_O_3_ column, precipitated in anhydrous methanol and dried under vacuum. (*l*-PS-PhN_3_, 93.4 mg, yield: 93.4%, *M*_n,SEC_ = 5000 g/mol, *M*_w_/*M*_n_ = 1.04).

### 2.6. Synthesis of Cyclic Double-Grafted Polymer (c-P2-g-Ph-PS)

Cyclic polymer (*c*-P2, 2.3 mg, 5 × 10^−4^ mmol), linear polymer (*l*-PS-PhN_3_, 81 mg, 1.62 × 10^−2^ mmol), PMDETA (5.62 mg, 3.24 × 10^−2^ mmol), mixture solvent (THF = 2 L, DMF = 1 mL) and a magnetic stirrer were added into a 10 mL ampoule in nitrogen atmosphere. CuBr (4.64 mg, 3.24 × 10^−2^ mmol) and Cu (1.72 mg, 2.7 × 10^−2^ mmol) were added into above ampoule. The solution was reacted at ambient temperature in nitrogen atmosphere. After 24 h, the polymer was precipitated in anhydrous methanol and dried in a vacuum. The crude double-grafted polymer was further purified by Prep-SEC to get final cyclic double-grafted polymer (*c*-P2-*g*-Ph-PS, 33.6 mg, yield: 40.33%, *M*_n,SEC_ = 30,700 g/mol, *M*_w_/*M*_n_ = 1.04).

## 3. Results and Discussion

### 3.1. Synthesis of l-PS-PhOH and l-PS-PhN_3_

Hydroxyl-containing polymeric double-chain (*l*-PS-PhOH) was synthesized by reacting 3,5-bis(propargyloxy)benzyl alcohol (Scheme 2) and azide-functionalized polystyrene (*l*-PS-N_3_) by virtue of Copper-catalyzed azide/alkyne cycloaddition (CuAAC) reaction. The synthesis and characterization of *l*-PS-N_3_ (*M*_n,SEC_ = 2500 g/mol, *M*_w_/*M*_n_ = 1.05) was shown in our previous publication [50]. The usage of slightly excessive *l*-PS-N_3_ was necessary in the process of preparing *l*-PS-PhOH, the gained crude *l*-PS-PhOH needs to be easily purified by Prep-SEC.

Hydroxyl-containing polymer double-chain (*l*-PS-PhOH) was verified by SEC, NMR, MALDI TOF MS and FT-IR spectroscopy. As shown in Figure 1, corresponding to *l*-PS-N_3_, ^1^H NMR spectra of *l*-PS-PhOH showed that the characteristic signal of the methine hydrogen (–CH(Ph)–, f) shifted from 3.8–4.1 ppm to 4.9–5.2 ppm completely. A new peak was clearly observed at 4.6 ppm, which was assigned to the benzylic hydrogen (–CH_2_–, i). In addition, the (f+h)/i/b integration ratio is close to 6/2/4, which means the successful formation of *l*-PS-PhOH. The number average molecular weight of *l*-PS-PhOH (*M*_n,SEC_ = 5000 g/mol, Figure 2) was twice than that of *l*-PS-N_3_ (*M*_n,SEC_ = 2500 g/mol, Figure 2) and the molecular weight distribution remained at 1.04, which also indicated the successful preparation of *l*-PS-PhOH. MALDI TOF MS (Figure 3) provided direct and powerful evidence for the formation of polymeric double-chain. The typical experimental peak *m/z* value (4715.97 Da) accords with the theoretical calculating value ([40M+Na]^+^, 4715.74 Da), in accordance with 40 repeat units of *l*-PS-PhOH with one sodium cation. The difference value of two adjacent experimental peaks is consistent with the *m/z* value of a styrene. Furthermore, in Figure 4, the complete disappearance of signals from azide groups (2094 cm^−1^) also proved the formation of *l*-PS-PhOH. All the above results confirmed the successful preparation of hydroxyl-containing polymer double-chain (*l*-PS-PhOH) without residue of *l*-PS-N_3_.

### 3.2. Synthesis of l-PS-PhN_3_ and c-P2-g-Ph-PS

Azide-containing polymer double-chain (*l*-PS-PhN_3_) was synthesized by hydroxyl-containing polymer double-chain (*l*-PS-PhOH) under the system of DPPA/DBU mixtures. Azide-containing polymer double-chain (*l*-PS-PhN_3_) was verified by SEC, NMR, MALDI TOF MS and FT-IR spectroscopy. Comparing to the spectrum of *l*-PS-PhOH (Figure 1), ^1^H NMR spectrum of *l*-PS-PhN_3_ (Figure 5) demonstrated that the benzylic hydrogen (–CH_2_–, i) shifted from 4.6 to 4.2 ppm completely, which indicated the complete formation of azide-containing polymeric double-chain (*l*-PS-PhN_3_). After azidation, the (f+h)/i/b integration ratio still kept at 6/2/4, which also indicated the successful preparation of *l*-PS-PhN_3_. From SEC curves (Figure 2 and Figure 6), there are no obvious changes before and after azidation. The average molecular weight of *l*-PS-PhN_3_ is 4900 g/mol and the molecular weight distribution remained at 1.03. In FT-IR spectrum (Figure 4), the vibrational absorption peak from azide group of *l*-PS-PhN_3_ appeared at 2094 cm^−1^. MALDI TOF MS provided persuasive evidence for successful formation of *l*-PS-PhN_3_. Figure 7 exhibited two main peak distributions, accurately assigned to polymeric double-chain (*l*-PS-PhN_3_). The typical experimental peak *m/z* value (such as 4712.91 Da), is consistent with the theoretical calculating value ([40 M-N_2_ + Na]^+^, 4712.74 Da). The difference value of two adjacent experimental peaks (103.94 Da) accords with the value of one styrene. These results demonstrated the successful preparation of *l*-PS-PhN_3_.

The active ester chemistry, such as the nucleophilic substitution of activated ester bearing pentafluorophenyl groups with diverse amines, is one kind of high-effective chemical reaction, which is often utilized as one post-modification technology for constructing functional polymers that cannot be obtained by conventional polymerization technologies [53]. Here, we chose propargylamine as the amine to react with cyclic poly(pentafluorophenyl 4-vinylbenzoate) (*c*-PPF4VB_6.5k_) for synthetizing functional cyclic polymer (*c*-P2). All the synthesis and characterizations of *c-*PPF4VB_6.5k_ and *c*-P2 are shown in Appendix A in detail (Appendix A).

Furthermore, functional cyclic polymer (*c*-P2) was used as cyclic polymeric backbone to react with polymer double-chain (*l*-PS-PhN_3_) via CuAAC reaction for constructing cyclic double-grafted polymer (*c*-P2-*g*-Ph-PS). The crude cyclic double-grafted polymer (*c*-P2-*g*-Ph-PS) was purified by Prep-SEC and further characterized by NMR and SEC. As shown in Figure 5, ^1^H NMR spectrum of *c*-P2-*g*-Ph-PS exhibited that the characteristic signals from the methine hydrogen (–CH(Ph)–, f) adjacent to 1,2,3-triazole, the methylene hydrogen (–CH_2_–, h and p) adjacent to 1,2,3-triazole and the benzylic hydrogen (–CH_2_–, i) were assigned in the 4.2–5.5 ppm region. It is hard to calculate grafting density of cyclic double-grafted polymer (*c*-P2-*g*-Ph-PS) by the integration ratio from ^1^H NMR spectrum, but the difference between *l*-PS-PhN_3_ and *c*-P2-*g*-Ph-PS indicated the successful preparation of cyclic double-grafted polymer. Additionally, according to the SEC curves of *c*-P2, *l*-PS-PhN_3_ and *c*-P2-*g*-Ph-PS (Figure 6), the shifts toward high molecular weight field can be observed clearly, demonstrating the successful formation of cyclic double-grafted polymer. The molecular weight of cyclic double-grafted polymer (*c*-P2-*g*-Ph-PS) was 30,700 g/mol and the molecular weight still stayed at 1.04.

## 4. Conclusions

A novel cyclic topological architecture, a cyclic double-grafted polymer, was successfully constructed using active ester chemistry and click chemistry via a “grafting onto” method. Cyclic backbone (*c*-P2) was synthesized by reacting propargyl amine with cyclic precursor (*c*-PPF4VB_6.5k_) using active ester chemistry as a post-modification approach. Hydroxyl-containing polymer double-chain (*l*-PS-PhOH) was prepared by reacting azide-functionalized polymer chain (*l*-PSN_3_) with 3,5-bis(propynyloxy)phenyl methanol, and further azide-modified to generate azide-containing polymer double-chain (*l*-PS-PhN_3_) and well characterized by SEC, NMR and MALDI TOF MS. Finally, this cyclic backbone (*c*-P2) was coupled with azide-containing polymer double-chain (*l*-PS-PhN_3_) using CuAAC reaction to successfully construct a novel cyclic double-grafted polymer (*c*-P2-*g*-Ph-PS). Notably, this cyclic double-grafted polymer (*c*-P2-*g*-Ph-PS) still exhibited a narrow molecular weight distribution. On the basis of our previous work, this research realized diversity and complexity of side chains from cyclic-grafted polymers, which could eventually enrich the topological architecture and provide a new platform for constructing amphiphilic cyclic-brush polymers with amphiphilic polymeric double-chains along the cyclic backbone.

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
