# Peer review of "Design and Synthesis of a Cyclic Double-Grafted Polymer Using Active Ester Chemistry and Click Chemistry via A “Grafting onto” Method"

_polymers, 2019, doi:10.3390/polym11020240_

Round 1

Reviewer 1 Report

In this work Authors reported about the synthesis of a cyclic double-grafted polymer, (c-P2-g-Ph-PS), applying a grafting onto approach. The proposed methodology does not represent a novelty, as evident from a recent article of the same Authors, (Polymer Chemistry, 2018, 9, 5155-5163). Nevertheless, I recommend the publication of the manuscript only after major revision, this because the introduction of a double-grafted polymer is  relatively new and adequately supported by experimental data.

1)      Figures 1, 2, and 4 referred to a comparison between l-PS-N3 and l-PS-PhOH data but l-PS-Br was indicated in each figure. Please correct with the exact abbreviation.

2)      On line 138 authors wrote that an excess of l-PS-N3 was necessary in the process of preparing –PS-PhOH. It is necessary explain the raison and add references.

3)      On line 196 two wrong terms are reported and are: trazide and traazide instead of triazole.

4)      The occurrence of the click reaction for obtaining c-P2-g-Ph-PS can be related to the appearance in the proton spectrum of a characteristic signal belonging to the triazole group. This is not reported by the authors that have to add this information to the manuscript and comment it. In figure 5 the second spectrum refers to c-P2-g-Ph-PS and not to c-PS-g-Ph-PS, please correct.

5)      When you talk about the characterization of c-P2-g-Ph-PS by NMR spectroscopy, you said that peculiar signals are observed at 4.2 – 5.5 ppm in the proton spectrum. This is not correct, no resonances are present in the 4.2-4.5 region , please adjust.

6)      Line 164 remove “and” after “synthesis of”

7)      Synthesis and characterization of c-PPF4VB6,5k and c-P2 have been already illustrated in a previous article of the Authors and must be removed from the supporting information paragraph. I also suggest to adopt the previously reported nomenclature  for PPF4VB and c-P.

8)      Line 87 were instead of was

9)      Line 125 polymer polymer, is a repetition. Correct with cyclic polymer c-P2 and linear polymer l-PS-PhN3.

10)   Line 128, the solution was kept at r.t. and not reacted.

11)   Authors must explain what are the potential applications of this polymer.

Author Response

Dear Reviewer 1,

Thank you very much for your constructive suggestions for our manuscript. We have revised our muscript according to your suggestions and upoload the a point-by-point response to you.

With my best regards,

Wei Zhang

---------------------------------------------------------------------------------------------

Wei Zhang, Ph.D, Prof.
Department of Polymer Science and Engineering
College of Chemistry, Chemical Engineering and Materials Science
Soochow University
TEL: +86-512-65884243 (office)
FAX: +86-512-65882787
EMAIL:
[email protected]

Reviewer 2 Report

The manuscript describes the synthesis and characterization of a cyclic poly(pentafluorophenyl 4-vinylbenzoate) grafted with polystyrene. It suggests an interesting synthetic route for a specialty polymer. I think that it is acceptable but some issues should be resolved before acceptance.

1.     In the introduction, the motivation of the synthesis should be specified. Why did authors synthesize the cyclic poly(pentafluorophenyl 4-vinylbenzoate) grafted with polystyrene? Is it a model polymer for specific applications? Even though applications as a biomaterial was mentioned, I can’t match the polymer with bioapplications. Also, how such a cyclic polymer with grafts are used for biomaterials? What do specific applications exist?

2.     Because the manuscript only focus on synthesis, it should provide more details in characterization for the following points

-       In line 64, the full name of c-PPF4VB6.5k should first appear before using abbreviation. Also, most of readers do not know what it is at first reading. Its chemical structure need to be inserted in Scheme 1 instead of providing in SI.

-       In materials and method part, provide city, state, country for all vendors.

-       Grafted polymer l-PSN3 is polystyrene as shown in the scheme, but the full name should be first appear in text before using the abbreviation.

-       At each section of material synthesis in the materials and method part, provide full 1H NMR and FT-IR data. Each proton should be indicated by using small alphabets, for example, Ha, Hb, … instead of ArH. Consider to place a chemical structure of each compound in each section of synthesis with the alphabets a, b, c,… for each proton.

-       Provide yield also for polymers.

-       In line 139, what does “to be slightly purified” means? How can a purification be slight?

-       In 1H-NMR spectra in Figure 1 and 5, some protons are missing in characterization. Those include aromatic and alcoholic protons in the phenyl ring including the proton i, a proton in the triazole ring, and protons in the methylene unit right next to the proton f.

-       In Figure 4 (1), l-PS-Br should be corrected to l-PS-N3.

-       In Figure 4 (1) and (3), I can see the existence of -OH band at around 3500 /cm. This means that the reaction might not be complete. Where is the OH peak in 1H NMR spectrum and does it completely disappear after reaction?  The proof of 1H NMR spectrum for a complete reaction should be provided. In this sense, the yield of the polymer synthesis should be provided.

-       Line 184-190 need to be mentioned in the introduction or materials and method part.

Author Response

Dear Reviewer 2,

Thank you very much for your constructive suggestions for our manuscript. We have revised our muscript according to your suggestions and upoload the a point-by-point response to you.

With my best regards,

Wei Zhang

---------------------------------------------------------------------------------------------

Wei Zhang, Ph.D, Prof.
Department of Polymer Science and Engineering
College of Chemistry, Chemical Engineering and Materials Science
Soochow University
TEL: +86-512-65884243 (office)
FAX: +86-512-65882787
EMAIL:
[email protected]

Reviewer 3 Report

         The paper by M. Liu et al. is well redacted and the conclusions are of great interest.

        A few remarks have to be taken in consideration :

        1- There are no values given concerning the characterisation of the products : 

             rate of crystallization, glass transition temperature, density for example .

         2- Some ideas concerning the possible use of these materials should be given.

         3- The synthesis of Cp2 with propargylamine is not referenced with (1) so the 

              discussion concerning the struture and the properties should be placed in 

               the article and not in the supporting information part .

Author Response

Dear Reviewer 3,

Thank you very much for your constructive suggestions for our manuscript. We have revised our muscript according to your suggestions and upoload the a point-by-point response to you.

With my best regards,

Wei Zhang

---------------------------------------------------------------------------------------------

Wei Zhang, Ph.D, Prof.
Department of Polymer Science and Engineering
College of Chemistry, Chemical Engineering and Materials Science
Soochow University
TEL: +86-512-65884243 (office)
FAX: +86-512-65882787
EMAIL:
[email protected]

Round 2

Reviewer 1 Report

The mauscript has been improved and can be published in the present form.

Reviewer 2 Report

Authors have sincerely responded all the issues and I think that the manuscript is acceptable to Polymers in the present form.